# COMPOSABLE PLANNING WITH ATTRIBUTES

## ABSTRACT

The tasks that an agent will need to solve often aren't known during training. However, if the agent knows which properties of the environment are important, then after learning how its actions affect those properties the agent may be able to use this knowledge to solve complex tasks without training specifically for them. Towards this end, we consider a setup in which an environment is augmented with a set of user defined attributes that parameterize the features of interest. We propose a method that learns a policy for transitioning between "nearby" sets of attributes, and maintains a graph of possible transitions. Given a task at test time that can be expressed in terms of a target set of attributes, and a current state, our model infers the attributes of the current state and searches over paths through attribute space to get a high level plan, and then uses its low level policy to execute the plan. We show in grid-world games and 3D block stacking that our model is able to generalize to longer, more complex tasks at test time even when it only sees short, simple tasks at train time.

## 1 INTRODUCTION

Deep reinforcement learning has demonstrated impressive successes in building agents that can solve difficult tasks, e.g. Mnih et al. (2015); Silver et al. (2016). However, these successes have mostly been confined to situations where it is possible to train a large number of times on a single known task or distribution of tasks. On the other hand, in some situations, the tasks of interest are not known at training time or are too complex to be completed by uninformed exploration on a sparse set of rewards. In these situations, it may be that the cost of the supervision required to identify the important features of the environment, or to describe the space of possible tasks within it, is not so onerous. Recently several papers have taken this approach, for example Reed & de Freitas (2015); Andreas et al. (2017); Oh et al. (2017); Denil et al. (2017).

If we expect an agent to be able to solve many different kinds of tasks, the representation of the task space is particularly important. In this paper, we impose structure on the task space through the use of *attribute sets*, a high-level abstraction of the environment state. The form of these are chosen by hand to capture task-relevant concepts, allowing both end goals as well as intermediate sub-tasks to be succinctly represented. As in Reed & de Freitas (2015); Andreas et al. (2017); Oh et al. (2017), we thus trade extra supervision for generalization.

The attributes yield a natural space in which to plan: instead of searching over possible sequences of actions, we instead search over attribute sets. Once the agent learns how its actions affect the environment in terms of its relevant attributes, novel tasks can be solved compositionally by executing a plan consisting of a sequence of transitions between abstract states defined by those attributes. In the experiments below, we will show that in various environments, training only on simple tasks, our agents are able to generalize to novel, more complex tasks.

## 2 MODEL

We consider an agent in a Markov environment, i.e. at each time the agent observes the state $s$ and takes action $a$, which uniquely determines the probability $P(s, a, s')$ of transitioning from $s$ to $s'$. We augment the environment with a map $f : S \to \{\rho\}$ from states to a set of user-defined attributes $\rho$. We assume that either $f$ is provided or a small set of hand-labeled $(s, \rho)$ pairs are provided in order to learning a mapping $\hat{f}$. Hence, the attributes are human defined and constitute a

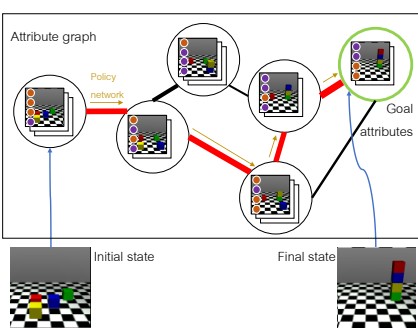

Figure 1: Solving complex tasks by planning in attribute space. Each state is mapped to a set of binary attributes (orange/purple dots). Our semi-parametric model comprises a graph over sets of attributes (e.g. "there is a blue block left of the red block"), with edge weightings according to the probability that a parametric policy network is able to transition between adjacent pairs. The attributes themselves are manually specified, but inferred from the observation through a neural network; and the graph structure and policy are learned during training via random exploration of the environment. Given a goal attribute set (green), we use the graph to find the shortest path (red) to it in attribute space. The policy network then executes the actions at each stage (gold arrows).

form of supervision. Here we consider attributes that are sets of binary vectors. These user-specified attributes parameterize the set of goals that can be specified at test time.

The model has three parts:

1. a neural-net based **attribute detector** $\hat{f}$, which maps states $s$ to a set of attributes $\rho$, i.e. $\rho = f(s)$.

2. a neural net-based **policy** $\pi(s, \rho_g)$ which takes a pair of inputs: the current state $s$ and attributes of the goal state $\rho_g$. Its output is a distribution over actions.

3. a tabular **transition function** $c_\pi(\rho_i, \rho_j)$ that scores the possibility of $\pi(s_{\rho_i}, \rho_j)$ transiting successfully from $\rho_i$ to $\rho_j$ in a small number of steps.

The transition table keeps track of the transitions seen in training, enabling a transition graph $G$ to be constructed that connects distant pairs of attributes. This high-level attribute graph is then searched at test time to find a path to the goal, with the policy network performing the low-level actions to transition between adjacent attributes.

Since our model uses attributes for planning, we desire a property that we will call "ignorability" which says that the probability of being able to transition from $\rho_i$ to $\rho_j$ should only depend on the attributes $\rho_i$, not the exact state; i.e. $P_\pi(f(s_{t'}) = \rho_j | f(s_t)) = P_\pi(f(s_{t'}) = \rho_j | s_t)$ [1]. To the extent that this condition is violated, then transitions are *aliased*, and a planned transition may not be achievable by the policy from the particular state $s$ even though it's achievable from other states with the same properties, causing the model to fail to achieve its goal. Note that in the experiments in 4.2, there will be nontrivial aliasing.

## 2.1 MODEL TRAINING

### 2.1.1 ATTRIBUTE DETECTOR $\hat{f}$

The first step of training consists of fitting the attribute detectors $\hat{f}$ that map states $s$ to attributes $\rho$. As mentioned above, we assume that there is a set of labeled (state, attribute) examples we can use for fitting this part of the model. Note that these examples are the *only* supervision given to the model during the entire training procedure.

### 2.1.2 ATTRIBUTE TRANSITION MODELS $c$ AND $c_\pi$

To construct $c$, the agent samples transitions from the environment, finding the sets of attributes which actually occur in the environment (from the potentially large number of all possible attributes). In the experiments below, we place the agent in a state at random with attributes $\rho_i$. It will then take a random action, or short sequence of actions. These lead to a new state with attributes $\rho_j$, and $c(\rho_i, \rho_j)$ is incremented. This is repeated many times, building up statistics on possible transitions within attribute space. The resulting table represents a transition graph $G$ with vertices given by all

---

[1]Note that the particular sequence of actions that effects the transition from $\rho_i$ to $\rho_j$ may still be conditional on the state.

the $\rho$ the agent has seen, and an edge between $\rho_i$ and $\rho_j$ counting the number of times the transition between $\rho_i$ and $\rho_j$ has occurred.

This procedure produces the graph $G$, but the counts in the graph are not normalized probabilities of transitioning from $\rho_i$ to $\rho_j$ (as we have never seen any negative samples). Therefore, given a low-level policy $\pi$ (see Sec. 2.1.3) we can optionally perform a second phase of training to learn the probability that for each $(\rho_i, \rho_j)$ in the graph, that if $\rho_i = f(s)$ then $\pi(s, \rho_i)$ will succeed at transitioning to $\rho_j$. At a random state $s$ with attributes $\rho_i$, we pick $\rho_j$ from the set of goals for which $c(\rho_i, \rho_j) > 0$ and see if $\pi$ is able to achieve this goal. While doing this, we keep track of the fraction of attempts for which the policy was successful at this transition and store this probability in $c_\pi(\rho_i, \rho_j)$.

### 2.1.3 Low level policy $\pi$

Finally, we need to train a policy network $\pi = \pi(s, \rho_g)$ to solve simple tasks, i.e. those that require a few actions to move between nearby attribute sets. One way of training $\pi$ is as an "inverse model" in the style of Agrawal et al. (2016); Andrychowicz et al. (2017). In the first phase of training the graph, suppose we sample each initial state $s_0$ and an action sequence $[a_0, a_1, ..., a_k]$ from the exploration policy, causing the agent to arrive at a new state with attributes $\rho_1$. We then treat $a_0$ as the "correct" action for $\pi(s_0, \rho_1)$ and update its parameters for this target. Alternatively, $\pi$ can be trained using reinforcement learning. After an initial graph $c$ is constructed, tasks can be chosen by sampling from states nearby the initial or current state properties.

## 2.2 Evaluating the model

Once the model has been built we can use it for planning. That is, given an input state $s$ and target set of attributes $\rho_T$, we find a path $[\rho_0, \rho_1, ..., \rho_m]$ on the graph $G$ with $\rho_0 = f(s)$ and $\rho_m = \rho_T$ maximizing

$$\sum_{i=0}^{m-1} \log c_\pi(\rho_i, \rho_{i+1}). \tag{1}$$

The optimal path can be found using Dijkstra's algorithm with a distance metric of $-\log(c_\pi(\rho_i, \rho_{i+1}))$. The policy is then used to move along the resulting path between attribute set, i.e. we take actions according to $a = \pi(s, \rho_1)$, then once $f(s) = \rho_1$, we change to $a = \pi(s, \rho_2)$ and so on. At each intermediate step, if the current attributes don't match the attributes on the computed path, then a new path is computed using the current attributes as a starting point (or, equivalently, the whole path is recomputed at each step).

## 3 Related work

**Hierarchical RL** Many researchers have recognized the importance of methods that can divide a MDP into subprocesses (Thrun & Schwartz, 1994; Parr & Russell, 1998; Sutton et al., 1999; Dietterich, 2000). Perhaps the most standard formalism today is the options framework of (Sutton et al., 1999), which deals with multistep "macro-actions" in the setting of reinforcement learning. Recent works, like Kulkarni et al. (2016), have shown how options can be used with function approximation via deep learning.

Our work is also a hierarchical approach to controlling an agent in a Markovian environment. However, the paradigm we consider differs from reinforcement learning: we consider a setup where *no reward or supervision is provided other than the* $(s, \rho(s))$ *pairs*, and show than an agent can learn to decompose a transition between far away $\rho, \rho'$ into a sequence of short transitions. If we were to frame the problem as HRL, considering each $\pi(\cdot, \rho)$ as a macro action[2], in order for the agent to learn to sequence the $\pi(\cdot, \rho_i)$, the environment would need to give reward for the completion of complex tasks, not just simple ones.

---

[2]Note also that all the "macro actions' in our examples in 4.2 are degenerate in the sense that they return after one step, but we still are able to show generalization to long trajectories from (unsupervised) training only on short ones

As opposed to e.g. Kulkarni et al. (2016), where additional human supervision is used to allow exploration in the face of extremely sparse rewards, our goal is to show that adding human supervision to parameterize the task space via attributes allows compositionality through planning.

**Horde and descendants** Our work is related to generalized value functions (Sutton et al., 2011) in that we have policies parameterized by state and target attributes. In particular, if we used a parameterized model for $c$, it would be similar to the factored state-goal representation in Schaul et al. (2015). Recently, van Seijen et al. (2017) used human provided attributes as a general value function (GVF) in Ms. Pacman, showing that using a weighted combination of these can lead to higher scores than standard rewards. Although the representation used in that work is similar to the one we use, the motivation in our work is to allow generalization to new tasks; and we use the attributes to plan, rather than just as tools for building a reactive policy.

**Factored MDP and Relational MDP** Our approach is closely related to factored MDP (Boutilier et al., 1995; 2000; Guestrin et al., 2003b). In these works, it is assumed that the environment can be represented by discrete attributes, and that transitions between the attributes by an action can be modeled as a Bayesian network. The value of each attribute after an action is postulated to depend in a known way on attributes from before the action. The present work differs from these in that the attributes do not determine the state and the dependency graph is not assumed to be known. More importantly, the focus in this work is on organizing the space of tasks through the attributes rather than being able to better plan a specific task; and in particular being able to generalize to new, more complex tasks at test time.

Our approach is also related to Relational MDP and Object Oriented MDP (Hernandez-Gardiol & Kaelbling, 2003; van Otterlo, 2005; Diuk et al., 2008; Abel et al., 2015), where states are described as a set of objects, each of which is an instantiation of canonical classes, and each instantiated object has a set of attributes. Our work is especially related to Guestrin et al. (2003a), where the aim is to show that by using a relational representation of an MDP, a policy from one domain can generalize to a new domain. However, in the current work, the attributes are taken directly as functions of the state, as opposed to defined for object classes, and we do not have any explicit encoding of how objects interact. The model is given some examples of various attributes, and builds a parameterized model that maps into the attributes.

The Programmable Agents of Denil et al. (2017) put the notions of objects and attributes (as in relational MDP) into an end-to-end differentiable neural architecture. Our work is similar to this one in that it includes learned mappings from states to attributes ($\hat{f}$ in our work, detectors in theirs; although we do not learn these end-to-end), and experiments in the setting of manipulating blocks in a physics simulator. On the other hand, our model uses explicit search instead of an end-to-end neural architecture to reason over attributes. Moreover, in Denil et al. (2017), the agent is trained and tested on similar tasks, but the object properties at test are novel; whereas our model is trained on simple tasks but generalizes to complex ones.

**Lifelong learning, multitask learning, and zero-shot learning**

There is a large literature on quickly adapting to a new learning problem given a set or a history of related learning problems. Our approach in this work shares ideas with the one in Isele et al. (2016), where tasks are augmented with descriptors and featurized. Our attributes correspond to these features. In that work, the coefficients of the task features in a sparse dictionary are used to weight a set of vectors defining the model for the associated task. In our work, the low level actor takes in the task features, but we learn how to transit between sets of features, and plan in that space. Similarly, the task is specified by a feature as an input into a model in Lopez-Paz & Ranzato (2017), again this corresponds to the way our low-level actor processes its goal.

Several recent deep reinforcement learning works have used modular architectures and hierarchy to achieve generalization to new tasks. For example, Tessler et al. (2017) uses pre-trained skills for transfer. Oh et al. (2017) uses a meta-controller that selects parameterized skills and analogical supervision on outer-product structured tasks. Our assignments of attributes serves a similar purpose to their analogical supervision, and we use parameterized skills as these works do. However, our "meta-controller" is the search over attributes, rather than a reactive model.

In Andreas et al. (2017), generalization is achieved through supervision in the form of "policy sketches", which are symbolic representations of the high level steps necessary to complete a given

task. The low level steps in executing modules in the sketches are composable. Our work is similar in that high level annotation is used to enable generalization, but the mechanism in this work is different. Note that the approaches in Andreas et al. (2017); Oh et al. (2017) are complementary to the one described here; in future work we wish to explore combining them.

**Semiparametric methods** In this work we use an explicit memory of sets of attributes the model has seen. Several previous works have used non-parametric memories for lowering the sample complexity of learning, e.g. Blundell et al. (2016); Pritzel et al. (2017). Like these, we lean on the fact that with a good representation of a state, it can be useful to memorize what to do in given situation (having only done it a small number of times) and explicitly look it up. In our case, the "good representation" is informed by the user-specified attributes.

Our approach is also related to Machado et al. (2017), which builds up a multiscale representation of an MDP using Eigenvectors of the transition matrix of the MDP, in the sense that we collect data on possible transitions between attributes in a first phase of training, and then use this knowledge at test time.

**Attributes in vision** Farhadi et al. (2009) and Lampert et al. (2009) explore visual attributes as a natural intermediate representation for object recognition tasks, demonstrating their effectiveness in one/low-shot settings. Subsequent work has applied the concept to fine-grained recognition (Duan et al., 2012), people's appearance/clothing (Zhang et al., 2014) and relative judgements (Parikh & Grauman, 2011). However, all these are static I.I.D settings, in contrast to dynamic agent/environment that this work explores.

## 4 EXPERIMENTS

We evaluate our approach (*Attribute Planner*, abbreviated to *AP*) in three different environments. The first two are randomly generated grid-worlds, and the third is a simulation of stacking blocks. In each environment, the goal of the model is to be able to generalize to testing on complex tasks from training on simpler tasks.

We compare against baseline policies trained in several ways. These baseline policies take the state and goal as inputs, and use the same neural network architecture as the policy used for the Attribute Planner.

1. **Reinforcement Learning:** Policies trained via reinforcement learning with A3C (Mnih et al., 2016) or Reinforce. We consider three variants of training: (i) training only with nearby goals (one attribute transition for grid-world; single actions for block stacking); (ii) training on the evaluation tasks; and (iii) training on a curriculum that transitions from nearby goals to evaluation tasks. Policies (ii) and (iii) are trained on full sequences, thus have an inherent advantage over our model, which only sees short sequences during training.

2. **Inverse:** An inverse model trained in a "supervised" fashion on a dataset of observed trajectories to predict the next action given the state and goal. We train on nearby goals and on longer multi-step tasks.

### 4.1 2-D WORLDS

We implemented two types of small 2-$D$ environments in Mazebase (Sukhbaatar et al., 2015), where the worlds are randomly generated for each episode. The action space for each consists of movements in the four cardinal directions, and additional environment specific actions.

**Colored Switches** The first environment consists of four switches, each with four possible colors. An extra toggle action cycles the color of a switch if the agent is standing on it. The attributes for this environment are the states of the switches; and the tasks are to change the switches into a specified configuration, as shown in Fig. 2(right). The locations and colors of the switches are randomly initialized for each episode.

**Crafting** In the second environment, similar to the one used in Andreas et al. (2017) an agent needs to collect resources and combine them to form items. In addition to moving in the cardinal directions, the agent has a "grab" action that allows it to pick up a resource from the current location

and add it to its inventory. If there is no item where the agent is standing, this action does nothing. The agent also has a "craft" action that combines a set of items to create a new item if the agent has the prerequisite items in its inventory and the agent is standing on a special square (a "crafting table") corresponding to the item to be crafted. If these two conditions are not both met, the "craft" action does nothing. The attributes for this environment are the items in the inventory, and task is to add a specified item to the inventory. In the environment, there are three types of resources and three types of products (see Fig. 2(left)). The episodes are initialized randomly by removing some resources, and adding some items to the inventory.

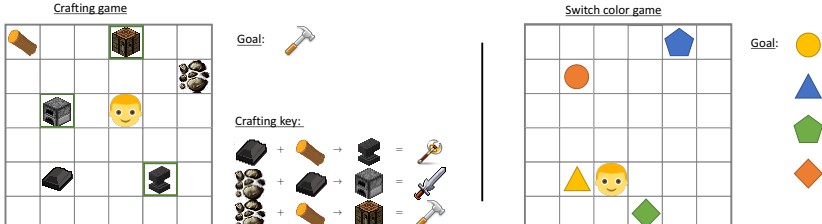

Figure 2: Left: Crafting mazebase game. Right: Colored switches game. See text for details.

The observation is given as a bag of words in both environments, where the words correspond to (feature, location). Features consist of item types, names, and their other properties. The locations include position relative to the agent in the maze, and also a few special slots for inventory, current, and target attributes.

The first phase building the high level transitions as in Section 2.1.2 is done by running a random agent in the environment from a random state until one or more attributes change. Then this change of attribute is recorded as an edge in graph $G$. The low level policy is trained concurrently with the second phase of Section 2.1.2. We use the current edge estimates in the graph to propose a target set of attributes, and the low level policy is trained with the Reinforce algorithm (Williams, 1992) to reach that set of attributes from the current state. These training episodes terminate when the task completes or after 80 steps; and a reward of -0.1 is given at every step to encourage the agent to complete the task quickly. The policy network is a fully connected network with two hidden layers of 100 units. We run each experiment three times with different random seeds, and report the mean success rate.

In the switches environment, multi-step (test) tasks are generated by setting a random attribute as target, which can require up to 12 attribute transitions. In the crafting environment, multi-step (test) tasks are generated by randomly selecting an item as a target. Since we do not care about other items in the inventory, the target state is underspecified. Some tasks are pre-solved because the randomly chosen target item can already be in the inventory. However, such tasks have no effect on training and are also removed during testing.

In curriculum training which we use as a baseline, we gradually increase the upper bound on the difficulty of tasks. In the switches environment, the difficulty corresponds to the number of toggles necessary for solving the task. The craft environment has two levels of difficulty: tasks can be completed by a single grab or craft action, and tasks that require multiple such actions.

During the second phase of training, we simultaneously compute the transition function $c_\pi(\rho_i, \rho_j)$ using an exponentially decaying average of success rates of the low level policy $\pi$:

$$c_\pi(\rho_i, \rho_j) = \frac{\sum_{t=1}^{T} \gamma^{T-t} M_\pi^t(\rho_i, \rho_j)}{\sum_{t=1}^{T} \gamma^{T-t} N_\pi^t(\rho_i, \rho_j)},$$

where $T$ is the number of training epochs, $N_\pi^t$ is the number of task $(\rho_i, \rho_j)$ during epoch $t$, and $M_\pi^t$ is the number of successful episodes among them. A decay rate of $\gamma = 0.9$ is used.

Table 1 compares our Attribute Planner (AP) model to a Reinforce baseline on the Crafting and Colored Switches tasks. In all cases, the baseline performs poorly, while the AP model has a high success rate. The importance of the graph is apparent: when it is removed (Reinforce method) the performance drops significantly on multi-step tasks.

| Method | Training data | Switches | | Crafting | |
|---|---|---|---|---|---|
| | | single | multi | single | multi |
| Reinforce | multi-step | 0.0% | 0.0% | 45.7% | 28.6% |
| Reinforce | multi-step + curriculum | 33.6% | 33.3% | 94.9% | 83.9% |
| Reinforce | one-step | **99.0%** | 15.4% | **98.7%** | 49.0% |
| AP | one-step | **99.0%** | **83.1%** | **98.7%** | **96.2%** |

Table 1: Task success rate on Mazebase environments. Our Attribute Planner (AP) approach performs much better than the reactive policies trained with Reinforce. The addition of the attribute graph (over Reinforce only model) is crucial for multi-step evaluation tasks.

## 4.2  STACKING BLOCKS

We consider a 3D block stacking task in the Mujoco environment Todorov et al. (2012). There are 4 blocks of different colors, and actions consist of dropping a block in a $3 \times 3$ grid of positions, resulting in 36 total actions. A block cannot be moved when it is underneath another block, so some actions have no effect. The input to the model is the observed image, and there are a total of 36 binary properties corresponding to the relative x and y positions of the blocks and whether blocks are stacked on one another. For example, one property corresponds to "blue is on top of yellow". Each training episode is initiated from a random initial state and lasts only one step, i.e. dropping a single block in a new location.

The policy network takes (i) a $128 \times 128$ image, which is featurized by a CNN with five convolutional layers and one fully connected (fc) layer to produce a 128d vector; and (ii) goal properties expressed as a 48d binary vector, which are transformed to a 128d vector. The two 128d vectors are concatenated and combined by two fc layers followed by softmax to produce an output distribution over actions. We use an exponential linear nonlinearity after each layer.

Table 2 compares the performance of different models on several block stacking tasks. In the *one-step* task, a goal is chosen that is the result of taking a single random action. In the *multi-step* task, the goal is chosen as the properties of a new random initialization. These tasks typically require $3 - 8$ steps to complete. In the *4-stack* task, the goal is a vertical stack of blocks in the order red, green, blue, yellow. We compare the performance of our model to reactive policy baselines trained on single-step tasks, complex multi-step tasks or with a curriculum of both. We perform each evaluation task 1000 times.

The single-step reactive policies perform well on single step tasks (which are what it sees at train time), but perform much worse compared to the AP model when transferred to multi-step tasks. The AP model without the second step of training that learns $c_\pi$ also performs substantially worse on multi-step tasks, demonstrating the importance of properly normalized transition probabilities to avoid aliased states.

The rightmost two columns of Table 2 consider underspecified goals, where only a subset of the attributes are provided. These are identical to their fully-specified counterparts, except that each attribute is left unspecified with probability 30%. The AP model handles these naturally by finding the shorted path to any satisfactory attribute set. We consider reactive baselines that are trained on the same distribution of underspecified attribute sets. Despite this, we observe that reactive policy performance degrades when goals are underspecified, while our AP model does not.

The attribute detector $\hat{f}$ predicts the full attribute set with $< 0.1\%$ error when trained on the full dataset of 1 million examples. If trained on only 10,000 examples, the attribute detector has an error rate of $1.4\%$. Training the AP model with this less-accurate attribute detector degrades multi-step performance by only 0.9%.

**Property Aliasing:** The "ignorability" assumption we made in Section 2 is violated in the block stacking task. To see why, consider a transition from "red left of blue and yellow" to "red right of blue and yellow". This can typically be accomplished in one step, but if blue and yellow are already on the far right, it cannot. Thus, states where this transition are possible and impossible are aliased with the same properties. This is the dominant source of errors on the multi-step task when trained on large sample sizes (in fact, it is the only source of errors as the policy approaches $100\%$

| Model | Training Data | one-step | multi-step | 4-stack | one-step | multi-step |
|---|---|---|---|---|---|---|
| | | | | | underspecified | |
| A3C | one-step | 98.5% | 8.1% | 1.9% | 65.7% | 6.6% |
| A3C | multi-step | 2.6% | 0% | 0% | 5.3% | 0% |
| A3C | curriculum | 98.2% | 17% | 2.9% | 8.2% | 0.2% |
| Inverse | one-step | **100%** | 9.1% | 0.5% | **98.8%** | 18.8% |
| Inverse | multi-step | 94.1% | 13.7% | 4.6% | 71.2% | 9.6% |
| AP (no $c_\pi$) | one-step | 74.5% | 29.7% | 62.2% | 81.8% | 28.1% |
| AP | one-step | 98.8% | **66.7%** | 98.5% | 97.8% | **63.5%** |

Table 2: Model comparison on block stacking task accuracy. Baselines marked 'multi-step' or 'curriculum' get to see complex multi-step tasks at train time. The Attribute Planner (AP) generalizes from one-step training to multi-step and underspecified tasks with high accuracy, while reinforcement learning and inverse model training do not. AP outperforms A3C even with a curriculum of tasks. Ablating the normalized graph transition table $c_\pi$ degrades AP performance substantially on multi-step tasks due to aliasing. Inverse one-step model was trained on 2 million examples, inverse multi-step and AP models were trained on 1 million examples, A3C models were trained to convergence.

| # of Training Examples | Inverse | | AP | |
|---|---|---|---|---|
| | one-step | multi-step | one-step | multi-step |
| 10,000 | 35.5% | 1.6% | 50.0% | 3.0% |
| 100,000 | 99.9% | 7.8% | 89.0% | 47.0% |
| 1,000,000 | 100% | 9.1% | 98.9% | 66.7% |
| 10,000,000 | 100% | 8.5% | 96.5% | 70.7% |

Table 3: Effect of the number of (one-step) training examples on one-step and multi-step performance, for an inverse model and the Attribute Planner model. The inverse models are trained on 2x the samples, including the samples generated from learning $c_\pi$ in our AP method.

accuracy and the graph becomes complete). Figure 4 shows an example plan that becomes stuck due to aliasing.

The second step of training, that computes the probability of $\pi$ transitioning on each edge, is important for mitigating the effects of aliasing in the block stacking task. The graph search finds the path with the highest probability of success (i.e. the product of probabilities on each edge), so it avoids edges that have high aliasing. In the AP model trained on one million samples, the second step of training improves multi-step performance from 29.7% to 66.7%, as shown in Table 2.

## 5 DISCUSSION

Our results show that structuring the space of tasks with high level attributes allows an agent to compose policies for the solutions of simple tasks into solutions of more complex tasks. The agent plans a path to the final goal at the level of the attributes, and executes the steps in this path with a reactive policy. Thus, supervision of an agent by labeling attributes can lead to generalization from simple tasks at train time to more complex tasks at test time. Nevertheless, there are many fronts for further work:

**Sample complexity of the planning module:** In Table 5 we can see both the benefits and the liabilities of the explicit non-parametric form for $c$. By 10K samples, the parametric lower level policy is already able to have a reasonable success rate. However, because in this environment, there are roughly 200K edges in the graph, most of the edges have not been seen, and without any weight-sharing, our model cannot estimate these transition probabilities. On the other hand, by 100K samples the model has seen enough of the graph to make nontrivial plans; and the non-parametric form of the graph makes planning straightforward. In future work, we hope to combine parametric models for $c$ with search to increase the sample efficiency of the planning module. Alternatively,

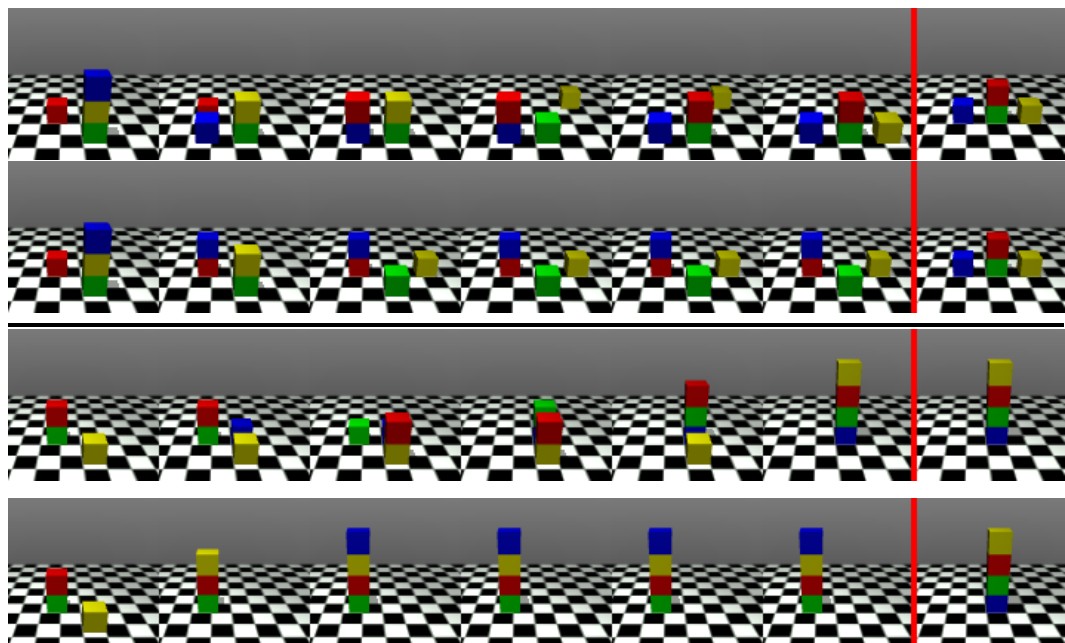

Figure 3: Two examples of block stacking evaluation tasks. The initial/target states are shown in the first/last columns. Successful completions of our Attribute Planner model are shown in rows 1 and 3. By contrast, the A3C baseline is unable to perform the tasks (rows 2 and 4).

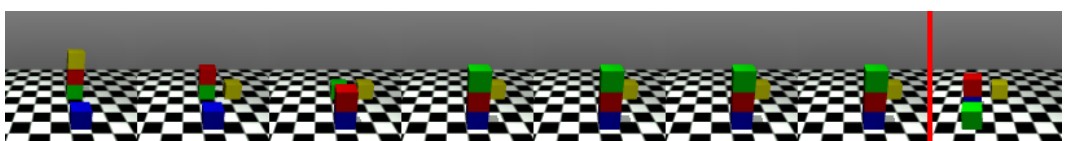

Figure 4: Plans become stuck when states with different transitions map to the same properties. In frame 4 of this example, the policy is directed to place the green block in front of the red and blue blocks, but this is impossible because the blue and red are already in the frontmost position.

we might hope to make progress on dynamic abstraction (projecting out some of the attributes) depending on the current state and goal, which would make the effective number of edges of the graph smaller.

**Exploration** Although we discuss an agent in an environment, we have elided many of the difficult problems of reinforcement learning. In particular, the environments considered in this work allow sampling low level transitions by starting at random states and following random policies, and these are sufficient to cover the state space, although we note that the method for training the model described in Section 2.1 allows for more sophisticated exploration policies. Thus we sidestep the exploration problem, one of the key difficulties of reinforcement learning. Nevertheless, building composable models even in this setting is nontrivial, and our view is that it is important to demonstrate success here (and decouple issues of exploration and composability) before moving on to the full RL problem.

We believe that the attributes $\rho$ and $c$, in addition to their usefulness for planning, provide a framework for incentivizing exploration. The agent can be rewarded for finding unseen (or rarely-seen) high level transitions, or for validating or falsifying hypotheses about the existence of entries of $c$.

**Learning the attributes:** Discovering the attributes automatically would remove much of the need for human supervision. Recent work, such as Thomas et al. (2017), demonstrates how this could be done. Another avenue for discovering attributes is to use a few "seed" attributes; and use aliasing as a signal that some attributes need to be refined.

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

## A    ADDING EXPLORATION

We also look into the use of exploration and learn a policy to promote exploration of undiscovered edges in our planning graph. We give a negative reward proportional to $\frac{t}{\sqrt{n}}$ where $n$ is the number of times that edge has been encountered before and $t$ the time episode. We compare this with a baseline method of single random start and random action selection over $N$ and the method used in the main paper, with $N$ random starts and single rollouts.

### A.1    MAZEBASE RESULTS

Our Mazebase environments in Section 4.1 are designed so that all edges can be discovered without much exploration. So to better test the benefit of exploration, we modified the craft environment to make discovering all edges harder. First, we added 4 new "super" products that can be crafted by combining one normal product with another resource, or three resources. Second, the environment always starts with only 3 resources and an empty inventory. Therefore, it is much harder to discover a super product because it requires agent to craft a product out of two resources, and then pick another resource and craft.

In this hard crafting environment, a random agent discovered 18.6 edges on average, while an agent with the exploration reward discovered all 25 edges of the environment. We used this complete graph to train our AP model and other baselines. Training on one-step tasks require episodes to start from different nodes of the graph, but the environment always initializes at the same node. A simple solution was not to reset the environment between episodes if the previous episode was successful. Thus the next episodes will start from a different node. Table 4 shows the success rates on multi-step tasks, where our AP model clearly outperforms the other baselines.

| Method | Training data | *Hard Crafting (multi-step)* |
|--------|---------------|------------------------------|
| Reinforce | multi-step + curriculum | 51.5% |
| Reinforce | one-step | 26.0% |
| AP | one-step | **99.8%** |

Table 4: Task success rate on multi-step task in the hard crafting environment. Our Attribute Planner clearly outperforms other baseline approaches.

### A.2    STACKING BLOCKS RESULTS

| # of Training | Policy | | Random | | Main Method | |
|---|---|---|---|---|---|---|
| Examples | edges | accuracy | edges | accuracy | edges | accuracy |
| 100,000 | 44k | 49.4% | 49k | 49.7% | 38k | 47.0% |
| 1,000,000 | 120k | 67.5% | 130k | 69.8% | 114k | 66.7% |
| 10,000,000 | 139k | 80.6% | 138k | 83.1% | 218k | 70.7% |

Table 5: Effect of the number of (one-step) training examples on one-step and multi-step performance, for an inverse model and the Attribute Planner model. The inverse models are trained on 2x the samples, including the samples generated from learning $c_\pi$ in our AP method.

