# OpenReview forum: "Composable Planning with Attributes"
_ICLR.cc/2018/Conference — Reject_

### Official Review · AnonReviewer2 · 2017-11-28
**Review for Composable Planning with Attributes**

**Rating:** 5
**Confidence:** 4

**Review:**

This paper proposed a method that enables hierarchical planning. Specifically, given human defined attributes, it learns a graph of attribute transitions. Given a test task and its target set of attributes and a current state, it infers the attribute of current state and search over paths through attribute spaces to get a high-level plan, and then use its low level policy to execute the plan.  Based on the relation (transition) of attributes, new and more complex tasks at test time can be solved compositionally. The proposed method is indeed technically sound and have some distinctions to other existing methods in literature, however, the novelty of this work does not seem to be significant as I will elaborate more.

1.	In this work, the attributes are provided by human, which certainty can incur a significant amount of effort hence limit is generalizable of the proposed method.  It might be more appealing if automatic attributes discovery can be incorporated into current framework to remove such restriction as well as better justify the assumption underlying the proposed method is that “the cost of the supervision required to identify the important features of an environment, or to describe the space of possible tasks within it, is not too expensive”

2.	the definition of ignorabilty a little confusing. “transition between \rho_i and \rho_j should only depend on the attributes \rho, not exact state” should be written.

3.	When evaluating the model, the authors mentioned that “We recompute the path at intermediate steps in case we reach an attribute set we don’t expect”. What does “attribute set we don’t expect” mean? Do you mean the attribute never seen before?

4.	The author should give better account of the relation between the proposed method to other frameworks. The authors mentioned that the proposed method can be placed into the framework of option. However, the option frame is mainly dealing with temporal abstraction, whereas this work seems have much more to do state abstraction.

5.	The current work is limited to dealing with problems with two levels of hierarchy

6. Minor comments
which properties of the environment we consider important -> which properties of the environment are important
a model that learns -> a method
from the set of goals rho_j -> from the set of goals,
GVF is undefined

---

> ### Comment · AnonReviewer2 · 2018-01-18
> **It is an iteresting paper, however I just have concerns about the significance of novelty**
>
> I just think the current work is not significantly novel,  given there are a few papers recently coming out with similar ideas at such as
> https://arxiv.org/pdf/1710.00459.pdf
> which also maps states to attributes (handcrafted) and solve hierarchical planning problems.

---

> > ### Comment · AnonReviewer1 · 2018-01-19
> > **Comment**
> >
> > I'm not sure that's quite fair to the authors, as the paper you linked to was only published on arXiv about three weeks before the ICLR deadline; I would consider that concurrent work.

---

### Official Review · AnonReviewer3 · 2017-11-28

**Rating:** 4
**Confidence:** 5

**Review:**

- This paper proposes a framework where the agent has access to a set of user defined attributes parametrizing features of interest. The agent learns a policy for transitioning between similar sets of attributes and given a test task, it can repurpose its attributes to reactively plan a policy to achieve the task. A grid world and tele-kinetically operated block stacking task is used to demonstrate the idea

- This framework is exactly the same as semi-MDPs (Precup, Sutton) and its several generalizations to function approximators as cited in the paper. The authors claim that the novelty is in using the framework for test generalization.

- So the main burden lies on experiments. I do not believe that the experiments alone demonstrate anything substantially new about semi-MDPs even within the deep RL setup. There is a lot of new vocabulary (e.g. sets of attributes) that is introduced, but it dosen't really add a new dimension to the setup. But I do believe in the general setup and I think its an important research direction. However the demonstrations are not strong enough yet and need further development. For instance automatically discovering attributes is the next big open question and authors allude to it.

- I want to encourage the authors to scale up their stacking setup in the most realistic way possible to develop this idea further. I am sure this will greatly improve the paper and open new directions of researchers.

---

> ### Comment · AnonReviewer3 · 2018-01-18
> **new comments**
>
> - The paper needs to be better contextualized with prior work. As other reviewers agree, the connections to MDPs and semi-MDPs is not really well crafted and compared.
>
> - As I mentioned earlier, the paper is promising and I wish to see it develop further. One productive avenue would be to scale the experimental setup in terms of visual richness or to enforce solving new stack configurations with one or few shot trial-by-error learning

---

### Official Review · AnonReviewer1 · 2017-11-28
**Impressive generalization from single-task training to multi-step planning**

**Rating:** 7
**Confidence:** 3

**Review:**

Summary: This paper proposes a method for planning which involves learning to detect high-level subgoals (called "attributes"), learning a transition model between subgoals, and then learning a policy for the low-level transitions between subgoals. The high-level task plan is not learned, but is computed using Dijkstra's algorithm. The benefit of this method (called the "Attribute Planner", or AP) is that it is able to generalize to tasks requiring multi-step plans after only training on tasks requiring single-step plans. The AP is compared against standard A3C baselines across a series of experiments in three different domains, showing impressive performance and demonstrating its generalization capability.

Pros:
- Impressive generalization results on multi-step planning problems.
- Nice combination of model-based planning for the high-level task plan with model-free RL for the low-level actions.

Cons:
- Attributes are handcrafted and pre-specified rather than being learned.
- Rather than learning an actual parameterized high-level transition model, a graph is built up out of experience, which requires a large sample complexity.
- No comparison to other hierarchical RL approaches.

Quality and Clarity:

This is a great paper. It is extremely well written and clear, includes a very thorough literature review (though it should probably also discuss [1]), takes a sensible approach to combining high- and low-level planning, and demonstrates significant improvements over A3C baselines when generalizing to more complex task plans. The experiments and domains seem reasonable (though the block-stacking domain would be more interesting if the action and state spaces weren't discrete) and the analysis is thorough.

While the paper in general is written very clearly, it would be helpful to the reader to include an algorithm for the AP.

Originality and Significance:

I am not an expert in hierarchical RL, but my understanding is that typically hierarchical RL approaches use high-level goals to make the task easier to learn in the first place, such as in tasks with long planning horizons (e.g. Montezuma's Revenge). The present work differs from this in that, as they state, "the goal of the model is to be able to generalize to testing on complex tasks from training on simpler tasks" (pg. 5). Most work I have seen does not explicitly test for this generalization capability, but this paper points out that it is important and worthwhile to test for.

It is difficult to say how much of an improvement this paper is on top of other related hierarchical RL works as there are no comparisons made. I think it would be worthwhile to include a comparison to other hierarchical RL architectures (such as [1] or [2]), as I expect they would perform better than the A3C baselines. I suspect that the AP would still have better generalization capabilities, but it is hard to know without seeing the results. That said, I still think that the contribution of the present paper stands on its own.

[1] Vezhnevets, A. S., Osindero, S., Schaul, T., Heess, N., Jaderberg, M., Silver, D., & Kavukcuoglu, K. (2017). FeUdal Networks for Hierarchical Reinforcement Learning. Retrieved from http://arxiv.org/abs/1703.01161
[2] Kulkarni, T. D., Narasimhan, K. R., Saeedi, A., & Tenenbaum, J. B. (2016). Hierarchical Deep Reinforcement Learning: Integrating Temporal Abstraction and Intrinsic Motivation. Advances in Neural Information Processing Systems.

---

> ### Comment · AnonReviewer1 · 2018-01-18
> **Still a good paper**
>
> After reading the other reviewers' comments and the authors' response, I still think this is a good paper and is worth publishing. First, I think the point about discovering attributes is important, but that it is out of scope for the present paper. Second, I agree the proposed framework is very similar to options*, but I think this ok: the novelty isn't so much in the framework itself but in that this is a very nice instantiation of it and that it demonstrates the efficacy of the compositionality afforded by options in more complex domains like block stacking.
>
> * I have to disagree with the authors when they say that their framework is different from options just because they are not doing RL. A MDP over options does not necessarily need to be solved with RL; if the transition function is known then you can use planning to solve the MDP which is exactly what the authors do here with Dijkstra's algorithm. I realize there is no explicit reward, but that doesn't mean the reward isn't implicit in the goal state. What's interesting about the present paper is that the agent has to also learn the high-level transition function through experience.

---

### Author Response · Authors · 2018-01-03
**Reply**

We thank the reviewers for their helpful feedback. We respond to the issues raised by R2 and R3 below. We also thank R1 and R2 for their detailed comments on clarity; we have amended the paper to address their feedback.


> “This framework is exactly the same as semi-MDPs (Precup, Sutton) and its several generalizations to function approximators as cited in the paper. The authors claim that the novelty is in using the framework for test generalization.”

We respectfully disagree.  The framework discussed in the paper is not a semi-MDP.  In our work, we have a Markovian environment which does not give any reward. Other than the attribute specification (either an explicit mapping or (state, attributes) training examples), all the agent’s interaction with the environment is unsupervised.

This is most visible in section 4.2 (Stacking Blocks).  In this experiment, the agent acts randomly at train time, and its policy is trained by trying to regress the action it actually took given a (state, next_attribute) pair.  This unsupervised training only sees 1-step transitions.  At test time, the agent is asked to go from a state to relatively distant set of attributes, requiring multiple steps.
Thus, while as mentioned in the text, there is a part of the setup that could be trivially framed in terms of options, the framing does not help with the tasks we study.

Fundamentally, semi-MDP and options are methods for formalizing temporal abstraction in an RL setting.   Our work is about using supervision to parameterize a space of tasks, independent of RL.  We have tried to explain that the learning paradigm described in the paper is not really even RL at all, even though we use some of the same language.  We will try to make the text clearer, but we also hope the reviewers can try to understand the learning paradigm we have studied, rather than trying to force our work into a hierarchical RL template.


> “No comparison to other hierarchical RL approaches”

As discussed, our paradigm differs from RL, thus it is not trivial to adapt hierarchical RL for this setting. In particular, hierarchical RL still requires a reward signal in order to learn a meta-policy over options, which is not provided in our setting; and in particular these methods require access to rewards from full-length trajectories, which our method does not.   Furthermore, most existing approaches to HRL require that a set of low-level options are provided (which they are not here);  there has been little success for methods that learn options in complicated state spaces with function approximation.


> “The author should give better account of the relation between the proposed method to other frameworks. The authors mentioned that the proposed method can be placed into the framework of option. However, the option frame is mainly dealing with temporal abstraction, whereas this work seems have much more to do state abstraction.”

We agree that the discussion of related work could be improved, particularly the connections to and differences with options frameworks. We will revise the paper to make this clearer.   However, note also that we emphasize in the related work that our approach is most similar to Factored MDP and Relational MDP; these deal with state abstraction.


> “In this work, the attributes are provided by human, which certainty can incur a significant amount of effort hence limit is generalizable of the proposed method.” “ For instance automatically discovering attributes is the next big open question and authors allude to it.”

We agree that automatically discovering attributes is an important and interesting problem, and would complement this work on using attributes for planning. However, we believe there is value in testing the building blocks of a system in isolation before trying to put the whole system together, and as learning attributes is challenging in its own right, it is not in the scope of this work.
We also believe that there are many situations where an attribute space can be readily provided externally, e.g. block stacking, starcraft, minecraft, house navigation, robotics, etc. Therefore, this approach has immediate value and doesn’t require the hard problem of representation learning to be solved. Furthermore, in the tasks we use for our experiments, we find that attribute specification is not overly onerous, requiring only a few thousand labeled examples to specify the attribute space for block stacking, and for grid worlds only a few hundred.

Other work, such as Policy Sketch (https://arxiv.org/pdf/1611.01796.pdf ; ICML best paper) also assumes that external task supervision can be provided without being learned from scratch.

---

### Decision · Program_Chairs · 2018-01-29
**ICLR 2018 Conference Acceptance Decision**

**Decision:**

Reject

**Comment:**

Overall the reviewers appear to like the ideas in this paper, though this is some disagreement about novelty (I agree with the reviewer who believes that the top-level search can very easily be interpreted as an MDP, making this very similar to SMDPs). The reviewers generally felt that the experimental results need to more closely compare with some existing techniques, even if they're not exactly for the same setting.